# The Effects of Irradiation on the Improvement in Oxidation Behavior of MX-ODS Steel in Liquid Pb

**DOI:** 10.3390/nano14090798

**Published:** 2024-05-04

**Authors:** Yuwen Xu, Shijing Xie, Jie Qiu, Cunfeng Yao, Wei Yan, Yanfen Li, Chongdou Yang, Shaoqiang Guo, Long Gu, Di Yun

**Affiliations:** 1School of Nuclear Science and Technology, Xi’an Jiaotong University, Xi’an 710049, Chinaguos2019@mail.xjtu.edu.cn (S.G.);; 2Institute of Modern Physics, Chinese Academy of Sciences, Lanzhou 730000, China; 3CAS Key Laboratory of Nuclear Materials and Safety Assessment, Institute of Metal Research, Chinese Academy of Sciences, Shenyang 110016, China; 4Shi-Changxu Innovation Center for Advanced Materials, Institute of Metal Research, Chinese Academy of Sciences, Shenyang 110016, China; 5State Key Laboratory of Multiphase Flow, Xi’an Jiaotong University, Xi’an 710049, China

**Keywords:** irradiation, ODS steel, liquid metal, corrosion

## Abstract

Lead-cooled fast reactors exhibit strong inherent safety performance and good economic features, while material degradation due to corrosion and irradiation is still challenging. Oxide dispersion-strengthened steels are one of the promising candidates for fuel cladding materials. The effects of both irradiation and corrosion on ODS steel need to be further studied. In this work, MX-ODS steel was irradiated by Fe ions at 500 °C up to 46 dpa. Later, the as-received specimen and the irradiated specimen were used to conduct corrosion tests in oxygen-saturated Pb at 550 °C for 1 h. In the as-received specimen, discontinuous oxides penetrated by Pb and Pb in contact with steel matrix were observed, demonstrating unsatisfactory corrosion resistance of the material. However, in the irradiated specimen after corrosion experiment, a protective oxide layer formed and prevented Pb attack. The oxidation behavior differences between the two specimens can be attributed to the defects produced by irradiation and the structural discrepancy in oxides caused by the formation process. A possible mechanism of irradiation on the corrosion is discussed. In the as-received specimen, Fe atoms loss led to voids in the oxides, and lead penetrated the oxides through these voids. In the irradiated specimen, defects left by previous irradiation helped to form a more uniform oxide layer. The adhesive outer magnetite oxide and the Fe ions generated from where grain boundary oxidation developed retarded the presence of voids and made the oxide layer protective.

## 1. Introduction

Lead-cooled fast reactors (LFRs) operating at high temperatures with a fast neutron spectrum exhibit strong inherent safety performance and good economic features. In LFRs, lead or lead–bismuth eutectic (LBE) is chosen as the coolant because of its low melting point, low viscosity, low chemical reactivity, high boiling temperature, low vapor pressure, etc. [1]. The safety of reactors is enhanced due to these favorable properties of the liquid metal coolant. However, these liquid metal coolants are highly corrosive. One of the major challenges for the development of LFRs is the corrosion of fuel cladding and structural materials, as the highly corrosive coolant sets a high requirement for the materials to withstand both intense neutron radiation and lead/LBE corrosion at the same time [2,3]. Therefore, under the conditions inside the reactors, material degradation due to the combined effects of corrosion and irradiation needs to be considered at a delicate level.

Oxide dispersion-strengthened (ODS) steels are one of the promising candidates for fuel cladding materials because of their outstanding high-temperature properties and remarkable tolerance to irradiation damage [4,5,6,7,8,9]. The corrosion behavior of various ODS steels has been tested and certain ODS steels showed good corrosion resistance owing to the formation of compact oxide layers on the surface. For example, Hosemann et al. studied the corrosion behavior of five ODS alloys and found that a continuous compact oxide layer formed at 535 °C could protect the materials from corrosion in LBE [10]. More studies revealed that the formation of dense oxides could retard the diffusion of atoms and present good corrosion resistance. This has been achieved by the addition of anti-corrosion elements such as Al [11,12,13,14] and Si [15] to ODS steels. Wang et al. [16] suggested that nano-oxides could act as nucleation sites, and the fine grain structure could provide a rapid diffusion channel for Al and Cr, improving corrosion resistance. With their corrosion behavior being studied, how irradiation affects the development of oxides and its impact on the corrosion behavior of ODS steels required further experiments and analysis.

The degradation of structural materials under irradiation and Pb/LBE corrosion conditions has been the focus of numerous studies in recent years. In situ irradiation and corrosion investigations have shown that irradiation could alter how materials corrode in liquid metal. Yao et al. [17] revealed that LBE corrosion of steel designed by the Institute of Modern Physics and the Institute of Metal Research, Chinese Academy of Sciences (SIMP steel) was accelerated by simultaneous heavy ion irradiation. The oxide layer became thicker, and its delamination and elementary composition were changed compared to the unirradiated region. In the in situ proton irradiation and LBE corrosion test of HT9 by Frazer et al. [18], oxides in the irradiated area were more than ten times thicker than those in the non-irradiated area due to the enhancement of diffusion by irradiation. Chen et al. [19] also observed thicker oxides in 11Cr1W1Si F/M steel in in situ proton irradiation and LBE corrosion. Lillard et al. [20] investigated the corrosion behavior of HT9 using electrochemical impedance spectroscopy and found that the corrosion rate of HT9 during proton irradiation immersed in LBE was increased due to the enhancement of oxygen vacancy concentration by irradiation. Additionally, LBE corrosion after heavy ion irradiation in Ref. [21] and Ref. [22] suggested that irradiation-induced defects resulted in more severe oxidation and spike-like oxides in the matrix.

With remarkable tolerance to irradiation damage of ODS steels, dispersive nanoparticles and fine grains could have effects on corrosion when irradiation is combined with corrosion. A novel MX-ODS steel (M = Ta/V, X = C/N) with extremely low C content was designed to service under high-temperature and radiation environments [23,24]. Understanding how irradiation affects the corrosion and oxidation behavior of MX-ODS steel in high-temperature liquid metal is important in ensuring its safe application in the LFRs systems. In this work, corrosion tests of the as-received and heavy-ion irradiated MX-ODS steel in the oxygen-saturated lead were conducted to probe the potential effects of irradiation on the corrosion behavior of MX-ODS steel. The corrosion properties of these two coupons were characterized and compared. The effect of irradiation on the corrosion behaviors of the ODS steel samples in liquid metal is discussed. It needs to be noted that in order to compare the corrosion behaviors of the tested materials with and without heavy-ion irradiation, only very short corrosion tests can be conducted. If the corrosion test time length becomes more extensive, then the corrosion behaviors will be dominated by the regions beyond the influence of irradiation and the comparison will be rendered ineffective. As a result, we conducted only 1 h corrosion tests on both the irradiated and unirradiated materials in order to clarify the effects of irradiation on the corrosion processes.

## 2. Material and Methods

### 2.1. Material

A newly developed MX-ODS steel with extremely low C content was used in this work, The MX-ODS steel was fabricated by powder metallurgy. The pre-alloy powder, smaller than 150 μm and produced by gas atomization, was mixed with 0.30 wt.% Y_2_O_3_ powder with a particle size of 100–200 nm. Mechanical alloying (MA) of the powder was conducted with a typical grinding device with a grinding rotation speed of 150 r/min for 90 h under the protection of high-purity Ar gas (99.99%). Then, the powder was consolidated by hot isostatic pressing (HIP) at 1150 °C with 170 MPa for 4 h, hot forging at 1200 °C for 3 h, and hot rolling at 1200 °C and 1000 °C. After hot rolling, the sheet material did not receive heat treatment. The nominal composition of the MX-ODS after hot rolling is shown in Table 1. There was no chemical loss after MA.

The precipitated phase in MX-ODS steel includes the MX phase of mainly VN, Y_2_O_3_ nanoparticles, and larger oxides containing various metallic elements. Most Y_2_O_3_ nanoparticles are under 10 nm and the oxide number density is about 5 × 10^22^/m^3^. MX-ODS steel contains no carbide due to its low C concentration. A detailed characterization of the MX-ODS steel can be found in previous work [23,24,25]. For the irradiation and subsequent liquid lead corrosion experiments, the as-received steel was cut into coupons with dimensions of 6 mm × 7 mm × 2 mm. Prior to irradiation, the coupons were ground down to 2000 grit with SiC paper to remove surface oxides and then polished with 0.5 μm alumina paste to mirror finish.

### 2.2. Irradiation Experiments

The ion irradiation experiment was carried out at the LEAF (Low Energy intense-highly-charged ion Accelerator Facility) facility at the Institute of Modern Physics, Chinese Academy of Sciences (IMP, CAS). Samples were irradiated by 19.6 MeV Fe ions at 500 °C for 57 h, to a total fluence of 5 × 10^16^ ions/cm^2^. A damage and injected ions profile was calculated by the Stopping and Range of Ions in Matter (SRIM) 2013 package under the quick Kinchin–Pease model [26]. The displacement threshold energy of Fe was set to 40 eV. As shown in Figure 1, irradiation influenced the bulk material from the surface down to a depth of 3.5 μm and peak damage of 46 dpa was obtained at the depth of 2.9 μm.

### 2.3. Corrosion Experiments

Liquid lead corrosion tests were conducted in a muffle furnace in an Ar glove box, where the oxygen concentration was controlled to be lower than 1 ppm. For both the as-received specimen and the irradiated specimen, the corrosion tests in Pb with saturated oxygen at 550 °C lasted for 1 h. Before the corrosion test, 1181 g solid lead was put into an alumina crucible and heated in the muffle furnace from room temperature to 550 °C at a rate of 5 °C/min. After reaching 550 °C, the samples, fixed in a holder made with aluminum oxide, were dipped into the oxygen-saturated liquid lead for the corrosion experiment. Note that, to attain saturated oxygen of about 10 ppm in the liquid lead at 550 °C [1], 0.014 wt.% PbO was added to the Pb used in corrosion tests. After 1 h exposure, the specimen was taken out of the liquid lead and was cooled in the furnace at a rate of 5 °C/min. Residual lead could be seen on the surface and the specimens were directly cold inlaid in epoxy for cross-sectional analysis without cleaning the surface.

### 2.4. Characterizations

After corrosion experiments, specimens were cold inlaid with epoxy, ground, polished, and then cleaned with acetone and ethyl alcohol for the cross-sectional examination. The cross-sectional morphologies and structures were characterized using scanning electron microscopy (SEM) and transmission electron microscopy (TEM). SEM analysis was carried out with a field emission scanning electron microscope (FE-SEM) Gemini SEM 500 (New York, NY, USA) equipped with an Oxford energy dispersive X-ray (EDS) spectroscopy system (Oxford, UK). All the cross-sectional SEM images were captured with a back-scattered electron detector (BSD). For further TEM analysis of corrosion specimens, focused ion beam (FIB) lift-out specimens were prepared in specific spots and Pt was used to protect the surface. TEM characterizations were completed with JEOL-F300 (Tokyo, Japan) and Talos-F200X (New York, NY, USA), with Oxford energy dispersive X-ray spectroscopy systems (Oxford, UK). The High-Angle Annular Dark Field (HAADF) detector was used to acquire dark field images.

## 3. Results

### 3.1. Effects of Irradiation

To confirm the effect of irradiation, TEM images and EDS maps were taken at different depths of the irradiated specimen and are shown in Figure 2. Figure 2a,b show the features in the depth below 4.5 μm, which is beyond the irradiation-affected area. The features are similar to the unirradiated specimen. No grain segregation of Cr was observed along the grain boundaries in Figure 2a. In Figure 2b, nano-oxide precipitates of Y_2_O_3_ with V and Cr shell structure could be seen in greater magnification. And insignificant grain boundary segregation of Cr with sizes smaller than 5 nm were observed, indicated by the arrow.

After Fe ion irradiation, as in Figure 2c,d, Cr enrichment at the grain boundary junctions as well as Cr and Mn grain boundary segregation were observed. Figure 2c was taken at a depth between 2 and 3 μm, where the damage level was 20~45 dpa. Obvious grain boundary segregation of Cr and Mn after irradiation was observed in this image. Figure 2d was taken at a depth of 250~500 nm, corresponding to a damage level of about 5 dpa. The images illustrate the Cr enrichment at the grain boundary junctions, leading to the formation of new phases rich in Cr and lack of Fe. These Cr-rich phases could be found throughout the irradiated area, independent of depth, i.e., dpa. In the whole specimen, no voids were observed due to the good swelling resistance of ODS steel. The size, composition, and distribution of nano-oxides in irradiated areas did not show differences compared to those in the as-received specimen. Nano-oxides were considered to be stable after irradiation.

### 3.2. Characterization of the As-Received Specimen after Corrosion

Figure 3 illustrates the results of cross-section SEM analysis of the as-received ODS steel after corrosion. Discontinuous oxides formed on the surface of the as-received specimen after being exposed in oxygen-saturated Pb at 550 °C for 1 h, and Pb was found in the oxides. Figure 3a shows the discrete oxides and Pb direct contact with the steel matrix, which was observed in most areas during cross-sectional analysis. In Figure 3b, the formation of an oxide scale could be seen and above is a layer of residual lead mixed with oxides. In Figure 3c, a zoomed-in image of the marked area in Figure 3b, Pb appeared beneath the oxides as marked. An EDS line scan through the oxides indicated the appearance of lead beneath the oxide layer, suggesting that this oxide layer could not protect against Pb penetration. In brief, the oxide layer formed only in part of the as-received specimen after corrosion, and could not prevent the steel matrix from direct contact with liquid lead.

To further confirm the property of the oxides, a FIB specimen was lifted out from a location similar to that in Figure 3c and was analyzed using TEM. As illustrated in Figure 4 and Figure 5a, an oxide layer containing mainly Fe, Cr and O was observed inside the original surface.

Fe oxides mixed with Pb could be seen outside the original surface, and the oxides were not continuous or adherent to the original surface. Thus, it was not recognized as the outer oxide layer. Furthermore, Pb penetrated the Fe, Cr oxide layer and came into contact with the steel matrix. Pb separated the steel matrix and the oxide and could lead to the further split of the oxides, which agrees well with the SEM cross-sectional analysis of Figure 3a. In Figure 5a,b, we presented both HAADF and bright field images of the same site in the oxide layer, with voids indicated by arrows. The grains of oxides were elongated and hundreds of nanometers in size. Voids were observed to sit between the oxide grains, as shown in Figure 5b with a 1000 nm under focus. An EDS point scan in Table 2 suggested that the atom ratio of Fe:Cr in the oxide was about 5:1. A selected area electron diffraction (SAED) pattern in Figure 6b indicates that the oxide scale is polycrystalline.

With EDS element analysis, the structure of the oxide was determined to be (Fe,Cr)_3_O_4_, commonly referred to as spinel. Notably, voids were observed throughout the oxides in the FIB specimen, also shown in Figure 5a. Additionally, some of the voids were filled with Pb, as clearly illustrated in a zoomed-in image in Figure 5b. Liquid lead penetrated the interface between the oxides and the steel matrix, filling the voids and degrading the oxide layer. The EDS results in Figure 6d,g show that lead firstly spread along the surface of the voids, and eventually filled them with lead. Figure 6e is a zoomed-in image of the voids in the oxides captured in a very thin area of the FIB specimen. The high-resolution image in Figure 6f shows the amorphous structure at the near edge area around voids.

### 3.3. Characterization of the Irradiated Specimen after Corrosion

Different from that of the unirradiated steel after corrosion, a thin and uniform oxide scale was observed in the irradiated specimen after corrosion, and no Pb penetration was found. As shown in Figure 7a, a uniform oxide film with an average thickness of around 0.7 μm was formed after 1h exposure in liquid Pb. A zoomed-in image in Figure 7b reveals that the oxide film is continuous and protective, with no Pb observed inside or beneath the oxides, as shown by the EDS line scan results. Additionally, oxidation along grain boundaries under the oxide films was observed in some areas due to the fast diffusion of oxygen along the grain boundaries, as indicated by arrows in Figure 7b.

A FIB lift-out specimen was prepared at a similar spot as in Figure 7b. A bi-layer oxide scale was observed in the irradiated specimen followed by the corrosion specimen, as shown in Figure 8. The outer layer contained Fe and O, and the inner part mainly contained Fe, Cr and O. The original surface, as marked by dashed lines, was recognized as the separation of the two oxide layers. The inner oxide layer was 300~500 nm in thickness, while the thickness of the outer layer could not be measured precisely since part of the oxides broke and fell during the preparation process of the SEM cross-section specimen. A bright field image in Figure 9a shows the morphology of oxides. The inner layer consisted of spherical grains of hundreds of nm and small grains of about 10 nm, while the outer layer was constituted of larger grains. A SAED pattern in Figure 9b with the point scan results in Table 2 indicated that the structure of the oxides were Fe_3_O_4_ and (Fe,Cr)_3_O_4_, which own similar structures and almost have the same d-spacings. They were often referred to as spinel and magnetite, respectively. The HAADF image in Figure 9c suggests that the inner layer consisting of small grains was not as compact as that of the larger grains. The SAED in Figure 9d and the high-resolution image in Figure 9e indicated the loss of crystalline structure around the small grains. nanomaterials-14-00798-t002_Table 2Table 2Results of EDS point scan in the oxides of the two specimens (sigma: standard deviation).
FeCrMnO
at%wt.%wt.% Sigmaat%wt.%wt.% Sigmaat%wt.%wt.% Sigmaat%wt.%wt.% SigmaA in Figure 5a27.0752.730.154.718.550.060.490.940.0467.7237.780.16B in Figure 9a25.0050.120.174.678.720.060.541.070.0469.7940.090.18C in Figure 9a32.0262.050.190.040.070.010.120.220.0267.8337.660.19


Additionally, oxidation along grain boundaries beneath the oxide layer was observed, as shown in Figure 8. Further EDS analyses of typical grain boundaries are presented in Figure 10 and Figure 11a. The oxides along grain boundaries showed higher content of Cr, and some also had higher Mn content, indicating that the oxides developed along segregation after irradiation, as revealed in Figure 10. The steel matrix adjacent to the grain boundary was also oxidized. Figure 11 shows a zoomed-in image where oxides of Cr and a small amount of Mn are present at the grain boundaries. Some of the oxides along the grain boundaries had a crystalline structure, while others were amorphous, indicated by circles, as shown in the high-resolution image in Figure 11b.

## 4. Discussion

### 4.1. Comparison of the As-Received Specimen and the Irradiated Specimen after Corrosion

The main idea of protecting the steel matrix from liquid metal attack or dissolution is traditionally pointing to the direction of the formation of a continuous oxide film [27,28]. In this work, the most significant difference observed in the oxides developed on the two specimens is their protectiveness against liquid lead, i.e., whether lead was observed inside the oxides or not. The as-received specimen was attacked by liquid lead, failing to form protective oxides, while a compact oxide film formed in the specimen which received ion irradiation, preventing the direct contact between lead and steel matrix, as shown in Figure 3 and Figure 7, respectively. Such results were unexpected since irradiation is regarded as having adverse effects on the lead/LBE corrosion performance of materials. The oxides in the irradiated specimen contained an inner layer of spinel and an outer layer of magnetite, which has been widely reported in oxygen-saturated Pb or LBE [29,30,31,32,33,34,35]. In the as-received specimen after corrosion, a large area suffered from Pb attack and dissolution, with oxides formed in part of the areas; however, they were not protective. In the irradiated specimen after corrosion, oxides were adherent and compact, with no lead observed in the oxides, while the oxides in the as-received specimen showed the existence of voids and were penetrated by liquid lead, indicating that the oxides were not compact during the oxidation process.

The distinct behavior of the as-received and irradiated specimens could be analyzed by the nature and formation process of the oxides in the two specimens. The grain morphologies of oxides were different in the two specimens: a mix of smaller grains and larger spherical grains in the irradiated specimen, and elongated grains in the as-received specimen. The atom ratio of Fe:Cr in the inner spinel of the irradiated specimen and the oxides in the as-received specimen was similar, at about 5:1, indicating that Cr remained and Fe diffused outward during oxidation in both specimens [36]. Since the two specimens were identical except for irradiation, and the conditions of corrosion were also identical, the oxides were basically the same as they were determined by thermodynamic conditions. It is the difference in the structure and formation process, rather than the composition, that had a stronger effect on their properties.

The distinctive corrosion behavior of the two specimens could be attributed to the differing formation processes of their oxides. A large number of defects induced by irradiation enhanced the diffusion of oxygen atoms, making oxygen atoms diffuse rapidly and uniformly through the grain lattice. Furthermore, the closely spaced defects acted as nucleation sites for oxide formation, reducing the distance required for the formation of a continuous oxide layer to cover the surface [37]. It should be noted that in this study, where corrosion tests were conducted after irradiation instead of an in situ experiment, defects generated by irradiation had a direct impact only on the initial stage of corrosion when oxygen diffused straightforward in the steel matrix. Once a continuous oxide layer formed, the oxidation rate was controlled by ions diffusion through this layer. While the effect of irradiation defects in steel matrix might be limited, the oxides generated in the irradiated specimen could not be considered the same as that in the non-irradiated specimen. In the corrosion experiment after irradiation in Refs. [21,22], a thicker oxide layer was observed in the irradiated specimen when a protective oxide layer formed in both specimens. This difference in thickness suggested that the oxide layer growing in the irradiated specimen had differences when allowing ions to pass through. Structural differences of oxides generated in the irradiated and as-received specimens were indeed observed in this work and will be discussed in later sections.

It should be emphasized that the MX-ODS steel used in this work was designed to obtain satisfactory mechanical properties at elevated temperatures and to endure irradiation damage. It was not designed to achieve good corrosion performance in liquid lead at high temperatures. The uneven oxidation could be due to the short duration of the experiment and the inhomogeneity of the steel, such as the orientations of the grains contact with liquid lead [38,39], especially considering that this MX-ODS steel demonstrated an average grain size of about 1 μm. Although the enhancement of diffusion by high-density grain boundaries could be considered as a method to improve corrosion resistance [40,41], the as-received specimen seemed to have not taken the advantage of small-size grains.

### 4.2. Grain Boundary Oxidation

Obvious segregation of Cr and Mn after irradiation was observed and led to oxidation at the grain boundaries. The segregation or depletion of Cr and other minor elements due to irradiation in ferritic/martensitic steels has been widely reported and could be the result of a combination of factors [42,43,44,45]. This work focuses on the oxidation behavior of irradiated and non-irradiated specimens; therefore, the segregation was not extensively investigated. The formation Gibbs energy of Cr and Mn oxide was smaller than that of Fe [46], meaning that Cr and Mn are more likely to form oxides. In the oxygen-saturated lead, sufficient oxygen was provided. Furthermore, grain boundary diffusion could be much faster than lattice diffusion [47]. Therefore, Cr and Mn enriched at grain boundaries combined with oxygen to form oxides. Oxidation at grain boundaries was reported in supercritical water by Chen et al. [48] and in liquid lead by Polekhina et al. [49], and were attributed to the fast diffusion of ions. Auinger et al. [50] suggested that the presence of Mn with its quickly changed valency states might allow faster ion transport at the oxides, also accounting for the prior grain boundary oxidation. Massive atoms diffused outward to accommodate for the oxides. The amorphous structure at the grain boundaries shown in Figure 11b could be the result of atoms diffusion.

### 4.3. Voids and Amorphous Structure in Oxides

Although oxides in both specimens were not perfectly compact, the irradiated specimen exhibited better resistance to liquid lead corrosion than the as-received specimen. The FIB specimen fabricated in the as-received specimen after corrosion, which showed oxides penetrated by lead, was analyzed to determine the causes of the fractured oxides mixed with lead on the steel surface. The main reason could be the presence of massive voids as shown in Figure 5. In addition to the obvious voids inside the oxides, there were possibly voids at the interface between oxides and matrix at the time of oxidation. However, the earlier oxidized interface was later filled with a mixture of oxides and lead, making it impossible to directly determine the morphology and properties of the voids.

The presence of voids in oxides has been reported in previous papers of liquid Pb/LBE corrosion tests of steels. The reason was attributed to rapid outward diffusion of Fe [28,51,52]. In this work, voids observed in the oxides could form via two mechanisms. On the one hand, atoms diffusion introduced vacancies that accumulated at the interface of oxides and matrix during the formation of oxides. The remaining space had not been occupied by the growth of oxides. As oxides grew inward, voids remained fixed and presented inside spinel. On the other hand, atoms within the oxides diffused outward into the liquid lead, particularly by the grain boundaries of oxides, leaving vacancies behind and eventually forming voids. With vacancies generated through both processes agglomerating, visible voids formed, deteriorating the compactness of oxides, also allowing the penetration of liquid lead. The penetration of lead into the oxides started from the inner surface of voids, as shown in Figure 6d,g, and eventually filled the voids. The loss of atoms led to a loose crystalline structure and amorphous areas were found around the voids, as illustrated in Figure 6f. The amorphous structure could be regarded as a result of atom loss, which was also found in the irradiated specimen after corrosion in Figure 9 and Figure 11.

In the irradiated specimen, no obvious voids were found in high number density within the oxide layer except for the amorphous structure around the small grains area in spinel. A similar loss of cations might have occurred in oxides of the irradiated specimen. Fortunately, the presence of the outer magnetite and the supplement of cations from grain boundary oxidation hindered this process. On the one hand, the presence of magnetite layer retarded the outward diffusion of Fe ions from spinel or from the steel matrix. On the other hand, due to the concentration gradient from the matrix to the oxide layer and then to the liquid lead, the atoms in the matrix tend to diffuse outwards. When grain boundary oxides developed, Fe atoms diffused outwards and left space for the formation of oxides. These outwards diffusion atoms could replenish the cation lost in the oxide layer to some extent and prevent the formation of voids.

### 4.4. Corrosion Process of the Oxides in Two Specimens

The corrosion process of the irradiated specimen and the as-received specimen was compared and illustrated in Figure 12.

In the irradiated specimen, mass defects, such as point defects and dislocations, were produced by irradiation, indicated by dots and lines in Figure 12. They enhanced the diffusion of oxygen inside the steel matrix. Oxygen diffused faster and was more uniformly even inside the grains, as shown in Figure 12a. A continuous and compact oxide layer formed, as illustrated in Figure 12b. The amorphous structures around small grains could be the result of Fe loss, which was retarded by the outer layer of magnetite and was replenished by Fe diffused from grain boundary oxidation beneath the oxide layer, as shown in Figure 12c. Small grains observed in the inner layer also provided fast diffusion paths for oxygen. The outward diffusion of Fe and inward diffusion of oxygen through the oxide layer led to the growth of double-layer oxides as in Figure 12d. The continuous oxides did not stop the process of oxidation but prevented the penetration of Pb into the steel matrix; therefore, no Pb attack was observed in the irradiated specimen after corrosion.

In the as-received specimen, in the beginning, oxides formed at the surface of the steel matrix unevenly. Oxides preferably formed at the area where oxygen could diffuse faster such as grain boundaries, and this led to a discontinuous oxide layer, as shown in Figure 12f. Compared to the irradiated specimen, oxygen inward diffusion in the matrix was slower without defects produced by irradiation. Fe diffusion outward left vacancies, which generated voids both inside the oxides and at the interface between oxides and steel matrix, as shown in Figure 12d. Oxides with voids were more fragile and not adhesive, and could be easily penetrated by liquid lead, as in Figure 12h. The compact oxide layer failed to form on the as-received specimen. Once Pb penetrated through the voids or the vulnerable points at the discontinuous oxides, the steel matrix came into direct contact with liquid lead. The oxides mixed with Pb might separate later. The lack of protection of the oxide may be due to its poor homogeneity. If the evenly formed oxide layer with no void forms during the whole corrosion process in a relatively low speed, the protection could last for longer time. As we considered irradiation defects as one of the reasons that led to the homogeneity and compactness of the oxide layer, defects during irradiation process may be utilized to improve the properties of the oxide layer.

## 5. Conclusions

Corrosion tests were conducted on the as-received and irradiated specimens of MX-ODS steel in oxygen-saturated Pb at 550 °C for 1 h. Fe ion irradiation up to 46 dpa at 500 °C caused grain boundary segregation of Cr and Mn. After the corrosion experiment, designed to avoid good corrosion resistance, the oxides of the as-received specimen were not uniform or protective, with penetration of Pb through the oxide layer. On the contrary, in the irradiated specimen after corrosion, a protective oxide layer was formed and no lead penetration was observed. The distinguishing behaviors of the as-received and irradiated specimens were analyzed by the nature and the formation processes of the oxides in the two specimens. In the as-received specimen, uneven diffusion of oxygen generated discontinuous oxides. With outward diffusion of Fe, vacancies accumulated inside the oxides or at the interface between the oxides and the matrix and generated voids, making the oxides vulnerable to Pb penetration. In the irradiated specimen, defects produced by irradiation enhanced the diffusion of oxygen into the grain lattice, leading to uniform oxides and protecting steel matrix from Pb attack. The oxidation of Cr and Mn at grain boundaries beneath the oxide layer was observed at some spots. Although there was atom loss indicated by the amorphous structure around the small grains in the spinel in the irradiated specimen after corrosion, the presence of outer magnetite and the Fe ions from where grain boundary oxidation developed mitigated the atom loss in the oxides. Therefore, there were no significant voids which would make the oxides vulnerable to liquid lead.

## Figures and Tables

**Figure 1 nanomaterials-14-00798-f001:**
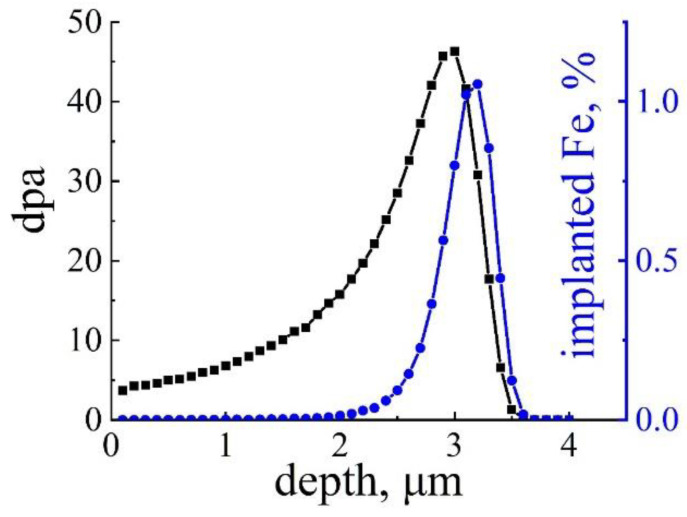
Damage and injected ions profile of 19.6 MeV Fe ion irradiation to total fluence of 5 × 10^16^ ions/cm^2^ calculated by SRIM 2013.

**Figure 2 nanomaterials-14-00798-f002:**
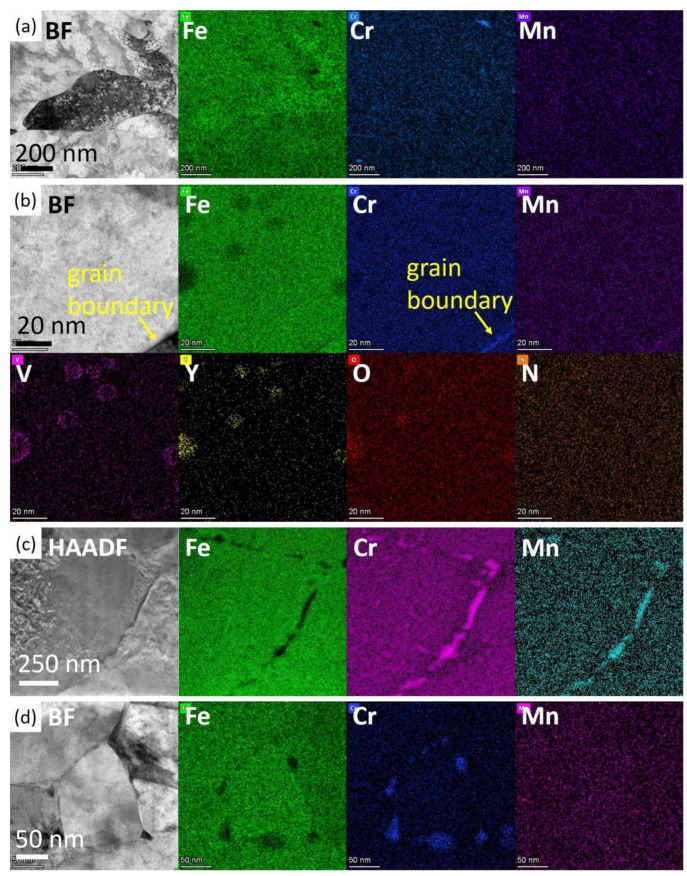
Grain boundary segregation without and with the effects of irradiation. (**a**,**b**) Images of the unirradiated areas and (**c**,**d**) images of irradiated areas. (**a**) No segregation or enrichment of Cr was seen without irradiation. (**b**) In greater magnification, Y_2_O_3_ nano-precipitates with V, and Cr shell and Cr enrichment at the grain boundary was smaller than 5 nm and could also be seen. (**c**) Obvious grain boundary segregation of Cr and Mn could be seen. (**d**) New phases rich in Cr formed at the intersection of grain boundaries.

**Figure 3 nanomaterials-14-00798-f003:**
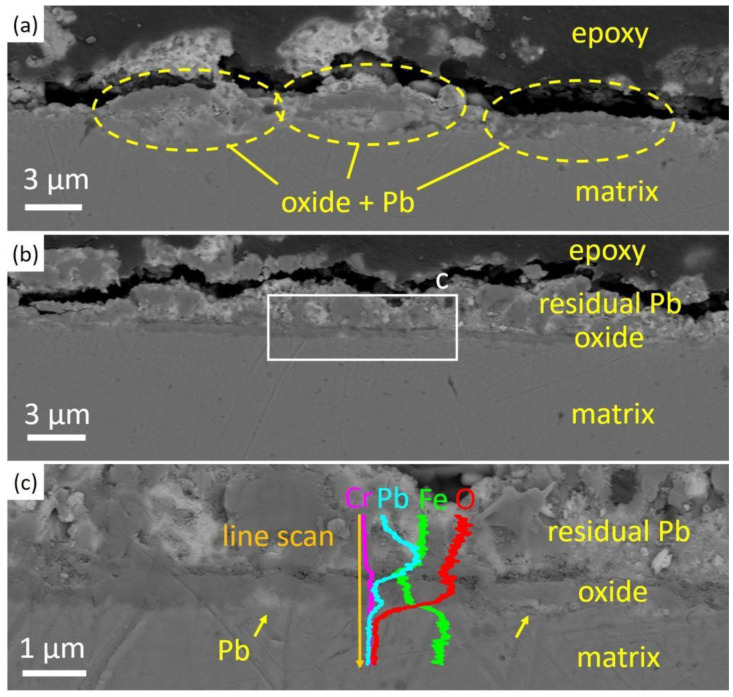
SEM cross-sectional images of the as-received specimen after Pb corrosion at 550 °C for 1 h. (**a**,**b**) Discontinuous oxides without protectiveness to the steel matrix. (**c**) A zoomed-in image of oxide layer. Pb penetration into the oxides are indicated with yellow arrows. EDS line scan results across the oxides show the presence of Pb inside the oxide layer.

**Figure 4 nanomaterials-14-00798-f004:**
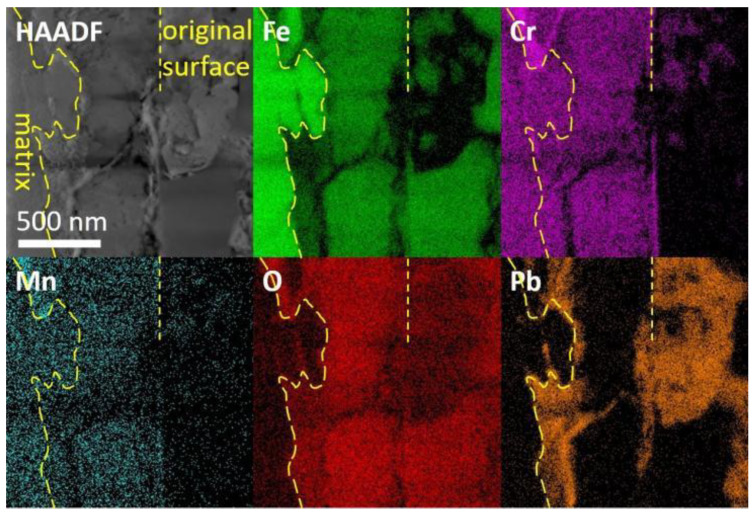
TEM and EDS images of the oxide layer in the as-received specimen after Pb corrosion. Fe and Cr oxides formed beneath the original surface. Pb was observed between oxides and steel matrix.

**Figure 5 nanomaterials-14-00798-f005:**
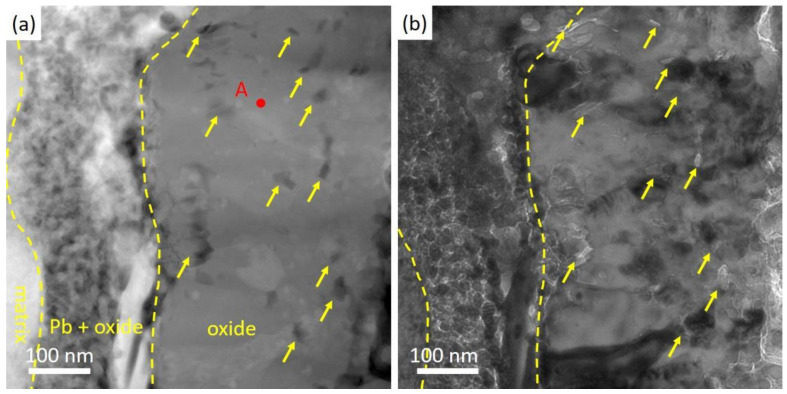
HAADF and bright field images of the oxide layer of as-received specimen after Pb corrosion. (**a**,**b**) HAADF and bright field images of the same spot, with arrows indicating the voids located between oxides grains. The bright field image was taken at 1000 nm under focus.

**Figure 6 nanomaterials-14-00798-f006:**
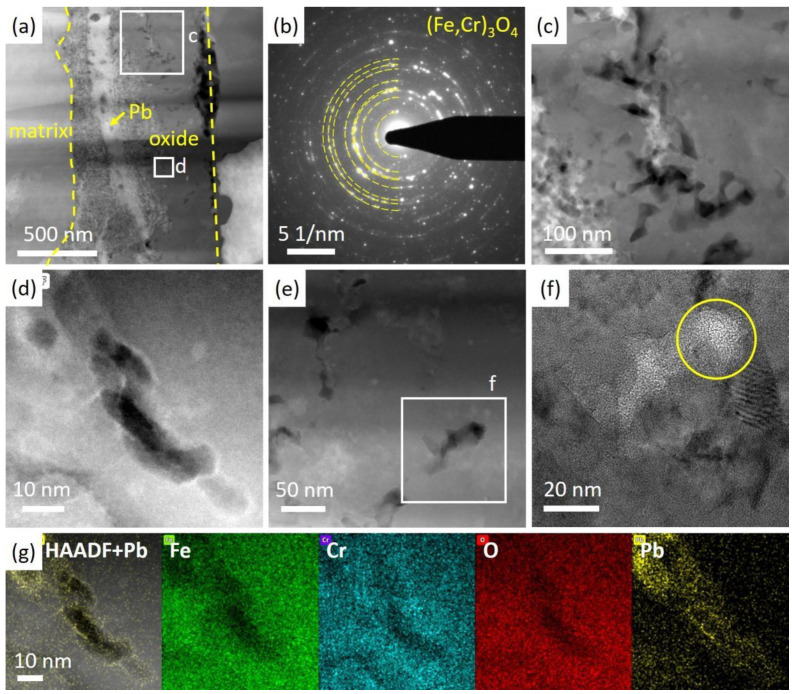
TEM images of the oxide layer in the as-received specimen after corrosion. (**a**) A HAADF image of the oxide layer. (**b**) The SEAD pattern of the oxides, revealing the structure of (Fe,Cr)_3_O_4_. (**c**) A zoomed-in image of voids with Pb filling in some of them. (**d**,**g**) A zoomed-in image of voids with Pb at the inner surface indicated by EDS results. (**e**) A HAADF image of voids captured in a very thin area in the as-received specimen after corrosion and (**f**) a high-resolution image in (**e**), showing the amorphous structure around the void as circled.

**Figure 7 nanomaterials-14-00798-f007:**
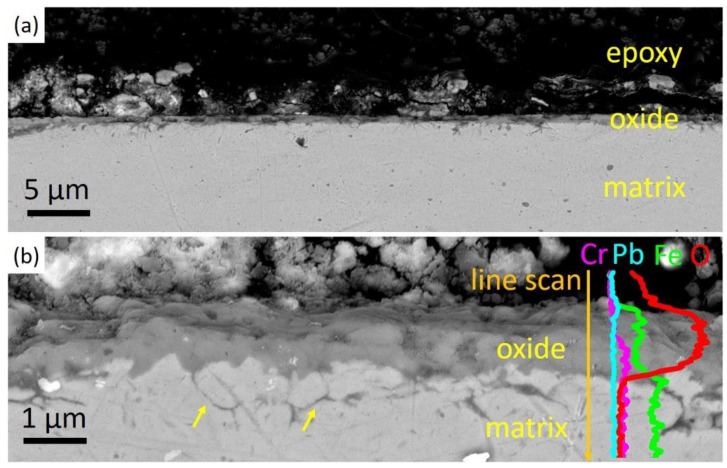
SEM cross-sectional images of the irradiated specimen after Pb corrosion at 550 °C for 1h. (**a**) Uniform oxide layer formed in the surface of steel matrix and no Pb penetration was observed. (**b**) A zoomed-in image of the oxide layer and its EDS line scan results. Oxidation along grain boundaries was observed in some areas indicated by arrows.

**Figure 8 nanomaterials-14-00798-f008:**
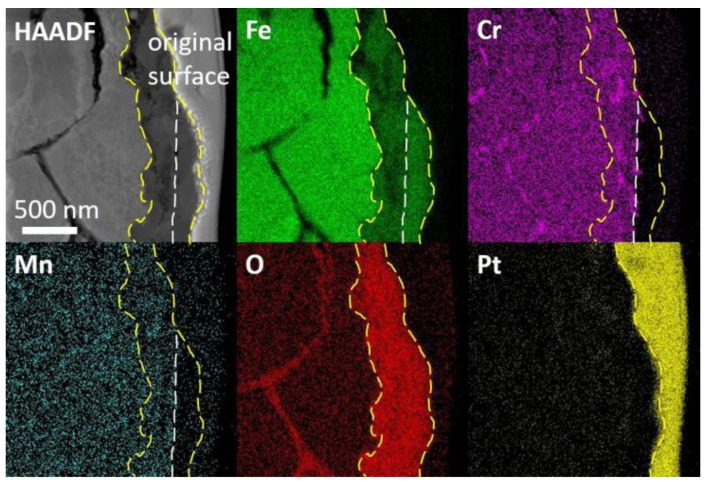
TEM and EDS images of the bi-layer oxide scale in the irradiated specimen after Pb corrosion. The original surface, marked by dash line, divided the outer layer and the inner layer.

**Figure 9 nanomaterials-14-00798-f009:**
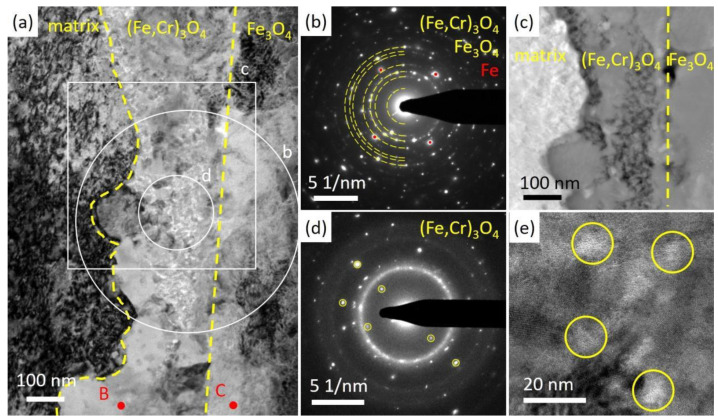
TEM images of the oxide layer in the irradiated specimen after corrosion. (**a**) Bright field image of the double-layer oxides in the irradiated specimen after corrosion. Both small and large grains could be seen in the inner layer. (**b**) SAED pattern of the oxides suggests the structure of Fe_3_O_4_ and (Fe,Cr)_3_O_4_. (**c**) HAADF image of the oxides. (**d**) SAED pattern of the small grains in the spinel suggests the loss of crystalline structure in this area. (**e**) High-resolution image of the small grains in spinel, amorphous area indicated by circles.

**Figure 10 nanomaterials-14-00798-f010:**
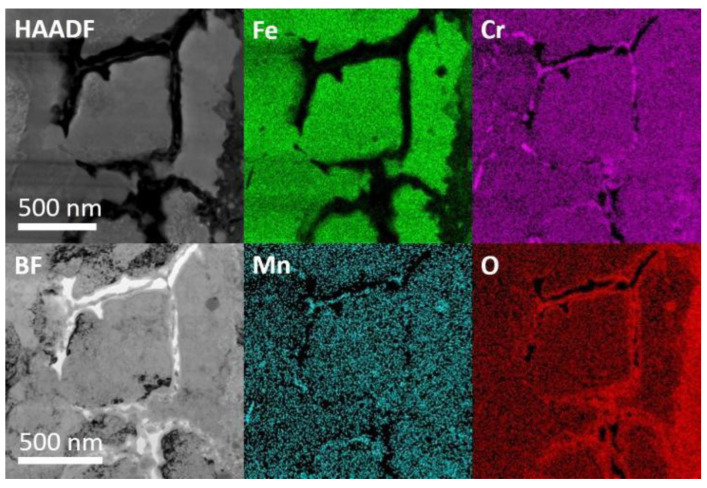
TEM and EDS images of the grain boundary oxidation with mainly Cr and Mn oxides beneath the oxide layer in the irradiated specimen after corrosion.

**Figure 11 nanomaterials-14-00798-f011:**
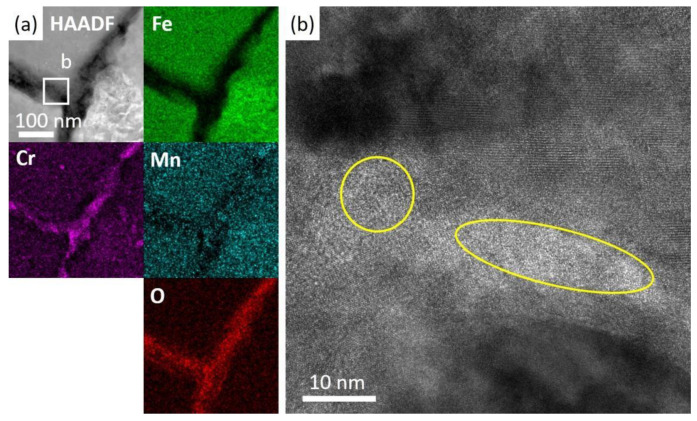
TEM and EDS images of grain boundaries in the irradiated specimen after corrosion. (**a**) Oxides of Cr and a small amount of Mn at the grain boundaries in the irradiated specimen after corrosion. (**b**) High-resolution image of the oxides. Some of the oxides had a crystalline structure while others were amorphous, indicated by circles.

**Figure 12 nanomaterials-14-00798-f012:**
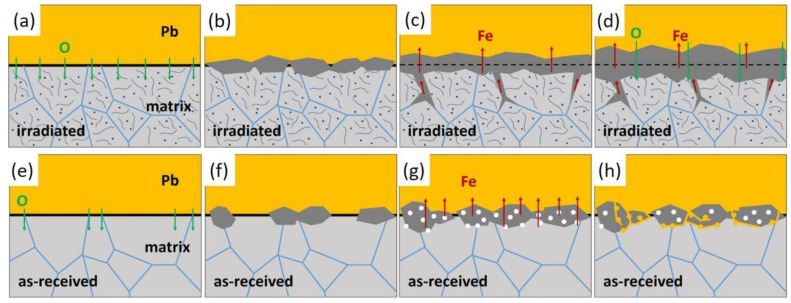
Schematic diagram for the corrosion process of the irradiated specimen in (**a**–**d**) and the as-received specimen in (**e**–**h**).

**Table 1 nanomaterials-14-00798-t001:** Nominal composition of MX-ODS steel used in this study (wt.%).

Fe	Cr	Mn	N	W	Ta	V	Y	O	C
Bal.	8.82	0.96	0.12	0.99	0.097	0.39	0.21	0.16	0.011

## Data Availability

The raw/processed data required to reproduce these findings cannot be shared at this time as the data also forms part of an ongoing study.

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
