# Peer review of "The Effects of Irradiation on the Improvement in Oxidation Behavior of MX-ODS Steel in Liquid Pb"

_nanomaterials, 2024, doi:10.3390/nano14090798_

Round 1

Reviewer 1 Report

Comments and Suggestions for Authors

In this manuscript the authors studied the effects of irradiation on the improvement of oxidation behavior of MX-ODS steel in liquid Pb. Overall, the manuscript is written well, and the authors conducted several analysis techniques to prove the concept of this work. However, to recommend this manuscript for publication in nanomaterials journal further improvement is required.

Comments:

1.      The abstract needs to be written again. Abstract should consist of: Introduction: the study's purpose and important background. Methods: A basic study design. Results: A summary of the important findings. Discussion topics include interpretations, findings, broader implications, and future research.

2.      All the abbreviations need to be defined before first use.

3.      How did the authors estimate the composition of MX-ODS steel (table.1). Also, please add the error range.

4.      Please correct the scale bar of TEM image figure 2.(d).

5.      Please error range in table 2. (SEM-EDS results).

6.      References formatting is wrong. Please correct it.

Comments on the Quality of English Language

Minor editing of English language required

Reviewer 2 Report

Comments and Suggestions for Authors

The nature of the defects produced by Fe ion irradiation is not clear. It is preferred to show a composite photograph wuth TEM multi-images over a wide area that shows the depth profile of irradiation defects corresponding to Figure 1. For the same reason it is not clear at which depth the defects are located in Figs. 2(c) and 2(d). In addition, as irradiation defects, dislocation loops and nano-precipitate such as YTiO or YAlO should have been observed because irradiation was performed at 500°C and these defect must be formed, but no detailed description of the microstructures is given here. This point also needs to be explained. Also, it is necessary to explain whether there was no formation of nano-precipitates.

Why is a compact oxide film formed by irradiation? Is it an enhancement of oxygen atom diffusion due to irradiation defects such as vacancies? I would like to know the reason why the oxide film is formed as a dense film.

the amount of corrosion produced by the corrosion test of irradiated material is more than that of the corrosion test under irradiation, and it is considered to be overestimated.

Intergranular corrosion will not occur only when the grain boundary properties are good, and ODS steel, which is formed by fine grains, is not a material with good corrosion resistance essentially.

The effect of irradiation on corrosion behavior in this work is not clear unless ion irradiation with higher energy of projectile particle is used. In materials with a defect concentration bias, the tendency to form vacancy and vacancy clusters such as voids seems to vary with depth. How do you consider the effect of this non-uniform defect formation due to depth profile of ion irradiation?

The movement of oxygen in oxides is considered to be basically unaffected by irradiation defects. Therefore, it is thought that there is no effect on the oxide with or without irradiation.

Consequently, we want to know whether irradiation under/before corrosion test improves or worsens the corrosion resistance. Is it a good thing that the accumulation of irradiation defects has homogenized the corrosion behavior on the surface? It could be considered that the homogenization of the oxide structure prevents attack by lead, but how does this affect the overall performance for corrosion resistance?

Comments on the Quality of English Language

Nothing special

Author Response

Please see the attachment. The reply is written in blue.

Reviewer 3 Report

Comments and Suggestions for Authors

This paper described the oxidation behavior of a newly developed MX-ODS steel with two statuses of as-received and irradiated. There are some problems for the authors to resolve, as follows:

1. L53, “SIMP”, define.

2. L80, what is the composition of the pre-alloyed powder before mixing with Y2O3? Are there any chemical losses after MA?

3. L86, how the sheet was cooled after hot rolling?

4. LL153-154, d), “Cr enrichment at grain boundary junctions as well as Cr and Mn grain boundary segregation were observed.” Why did Cr and Mn segregate at the grain boundary following irradiation?

5. L159, “These Cr-rich phases could be found…” Are there any changes of the Y2O3-V nano-oxide precipitates as an ODS steel? What is the function of these nano-oxides?

6. LL196-197, “The grains of oxides were elongated and hundreds of nanometers in size.” why the grains were elongated?

7. LL258-260, “The oxides along grain boundaries showed higher content of Cr, and some also had higher Mn content, indicating that the oxides developed along segregation after irradiation, as revealed in Fig. 10.” Is that cracks at the grain boundary in Figure 10, some black areas at the grain boundaries are not indexed?

8. L366, “The amorphous structure could be regarded as a result of atom loss,…” Is the amorphous area also the Cr-Mn-rich area? meaning the loss of Fe due to oxidation? What is the oxidation sequence of elements, including Fe, Cr and Mn?

9. L379 in section 4.4, what is the difference between the as-received and irradiated samples? It can only be seen that the irradiated sample shows higher percentages of Cr-Mn segregations at the grain boundaries. While in figure 12, there are more features for irradiated sample within the grains. what is that?  Still the question, for the ODS steel, what is the function of the nano particles (V-Y-O) here?

Comments on the Quality of English Language

understandable
